# The Impact of Non-Donor-Specific HLA Antibodies on Antibody-Mediated Rejection in Pediatric Kidney Transplant Recipients

**DOI:** 10.3390/ijms26125870

**Published:** 2025-06-19

**Authors:** Maria Sangermano, Vittoria Soncin, Maria Auciello, Francesco Ciabattoni, Susanna Negrisolo, Elena Marinelli, Nicola Bertazza Partigiani, Elisa Benetti

**Affiliations:** 1Pediatric Nephrology, Department of Women’s and Children’s Health, Padua University Hospital, 35128 Padua, Italy; maria.sangermano@aopd.veneto.it (M.S.); vittoria.soncin@aopd.veneto.it (V.S.); maria.auciello@aopd.veneto.it (M.A.); francesco.ciabattoni@aopd.veneto.it (F.C.); elena.marinelli@aopd.veneto.it (E.M.); nicola.bertazzapartigiani@aopd.veneto.it (N.B.P.); 2Laboratory of Immunopathology and Molecular Biology of the Kidney, Department of Women’s and Children’s Health, Padua University Hospital, 35128 Padua, Italy; susanna.negrisolo@unipd.it

**Keywords:** antibody-mediated rejection, mean fluorescence intensity, non-donor-specific antibodies, pediatric kidney transplantation, viral infections

## Abstract

While the pathogenic role of donor-specific anti-HLA antibodies (DSAs) in long-term immune-mediated injury after kidney transplantation is well established, the clinical relevance of non-donor-specific antibodies (nDSAs), also detected in transplant recipients, remains a subject of debate. This retrospective study evaluated the prognostic value of nDSAs in 92 pediatric kidney transplant recipients (89.1%, 9.8%, and 1.1% for first, second, and third transplants, respectively) at the University Hospital of Padua between January 2015 and December 2022, investigating the association between antibody development and clinical outcomes, including graft function, rejection episodes, and viral infections. Clinical, immunological, virological, and histopathological data were collected at 6, 12, and 24 months post-transplant. Antibody prevalence increased over time, with nDSAs being more frequent than DSAs at all timepoints. The combined presence of DSAs and nDSAs significantly increased the risk of ABMR (HR = 45.10; *p* < 0.001). Isolated nDSAs and DSAs were also associated with an increased risk of ABMR (HR = 6.43 and 12.10, respectively), suggesting a synergistic alloimmune effect. Viral infections also emerged as relevant cofactors in humoral alloimmunity. EBV viremia and intrarenal Parvovirus B19 (PVB19) infection were significantly associated with ABMR, with PVB19 also correlating with nDSA formation. In conclusion, integrated immunological and virological monitoring may support risk stratification and guide individualized post-transplant management. Larger multicenter studies are warranted to define the long-term impact of nDSAs in pediatric kidney transplantation.

## 1. Introduction

Kidney transplantation has markedly improved survival and quality of life for children with end-stage renal disease, yet long-term allograft outcomes remain suboptimal. Despite declining rates of early acute rejection, late allograft failure persists as a major challenge, with antibody-mediated rejection (ABMR) identified as one of the most frequent causes of late graft loss [1]. ABMR is primarily driven by donor-specific anti-HLA antibodies (DSAs) directed against the graft. It is widely accepted that the presence of HLA DSAs correlates with endothelial injury, complement activation, and a substantially elevated risk of ABMR and graft failure [2,3]. Pediatric recipients are not exempt: multicenter analyses report ABMR incidences of approximately 10–20% within the first 5 years post-transplant [4].

Pediatric kidney transplant recipients exhibit distinct immunological and virological characteristics that influence their susceptibility to HLA antibody development. Factors such as young age, immunological naivety, a highly reactive immune system, and frequent exposure to primary viral infections (e.g., CMV, EBV, and BKPyV) contribute to heightened immunologic risk [5]. Moreover, the long-term nature of pediatric transplantation, with a high likelihood of re-transplantation and cumulative immune insults over time and adolescent non-adherence therapy, further complicates immune monitoring and management [6,7]. These findings support the utility of post-transplant HLA antibody surveillance as a tool to identify early immunologic activity that may precede graft dysfunction, even in patients initially considered at low immunological risk.

While the harmful effects of donor-specific anti-HLA antibodies (DSAs) on graft survival are well established, many transplant recipients also develop non-donor-specific anti-HLA antibodies (nDSAs), whose clinical relevance—especially in pediatric transplantation—remains unclear. In adults, some studies have reported an association between nDSAs and poorer graft outcomes, including reduced renal function and survival [8,9], but the results are heterogeneous and not directly transferable to pediatric populations. In one study of previously unsensitized children, over 30% developed de novo nDSAs within 1–2 years after transplant, yet these antibodies were generally low-strength and non-complement-fixing and did not induce detectable graft injury in the medium term [10]. Thus, while DSAs are recognized markers of alloimmune risk, the prognostic significance of nDSAs in pediatric kidney transplantation remains poorly defined and merits further investigation.

In light of these knowledge gaps and the discordant findings reported in the literature, we conducted a retrospective study to better characterize the emergence and clinical impact of anti-HLA antibodies in a pediatric transplant population, aiming to contribute to the understanding of the immunological dynamics of nDSAs in pediatric kidney transplantation and to clarify their potential prognostic value. In particular, we aimed to assess the frequency of de novo DSA and nDSA formation after transplantation through a systematic analysis of sequential serum samples, to characterize the antigenic targets and functional profile of the detected antibodies, and to investigate the association between antibody development and clinical outcomes such as graft function, rejection episodes, and viral infections.

## 2. Results

### 2.1. Population Characteristics

A total of 92 pediatric kidney transplant recipients were included in this study. The median age at transplantation was 11.5 years (interquartile range [IQR]: 9.25). Males represented 64.1% of the cohort. Most patients (89.1%) underwent their first transplant, whereas 9.8% received a second, and 1 (1.1%) a third. A total of 80.4% of children had received dialysis pretransplant (peritoneal dialysis in 41.3%, hemodialysis in 22.8%, and a combination of both in 16.3% of children). The most frequent underlying cause of end-stage renal disease (ESRD) was congenital anomalies of the kidney and urinary tract (CAKUT) in 47.8%, followed by glomerulopathies (20.6%) and ciliopathies (16.3%). The mean HLA mismatch at transplant was 3.57 ± 1.31, with 25.0% of recipients at a mismatch level of three. Immunosuppressive regimens included basiliximab for 85.9% of the patients, while the remaining 14.1% received ATG. Maintenance therapy commonly comprised tacrolimus (63.2%) and mycophenolate mofetil in all patients. All patients maintained immunosuppressive drug concentrations within the recommended therapeutic ranges. Baseline nDSAs were detected in 38.0% of patients: 21% had class I nDSAs, 17% had class II nDSAs, and 10% had both classes. No patient presented DSAs at the time of transplantation. Population characteristics are presented in Table 1.

### 2.2. Kidney Function and Proteinuria over Time

No patients died or presented graft failure during the follow-up period. Kidney function assessed by the eGFR at 6, 12, and 24 months post-transplant demonstrates an eGFR median value of 75.0 mL/min/1.73 m^2^ (IQR 22.5) at 6 months, 76.0 mL/min/1.73 m^2^ (IQR 31.5) at 12 months, and 72.0 mL/min/1.73 m^2^ (IQR 33.0) at 24 months. The Friedman test showed no significant change over time (χ^2^ = 0.0762, degrees of freedom [df] = 2, *p* = 0.963). Proteinuria had a median value of 163 mg/m^2^/24 h (IQR 183 mg/m^2^/24 h) at 6 months, 147 mg/m^2^/24 h (IQR 128 mg/m^2^/24 h) at 12 months, and further increased to 201 mg/m^2^/24 h (IQR 192 mg/m^2^/24 h) at 24 months. The Friedman test revealed a significant rise over the three timepoints (χ^2^ = 8.68, df = 2, *p* = 0.013) (Figure 1). Furthermore, baseline nDSAs positivity correlated with reduced pretransplant eGFR (r = −0.314, *p* < 0.001) and, to a lesser extent, with higher proteinuria (r = 0.151, *p* = 0.012).

Analysis by ABMR status indicated that eGFR at 6 months was lower in patients who developed ABMR (median 67 vs. 79 mL/min/1.73 m^2^; *p* = 0.042), with a borderline difference at 12 months (*p* = 0.071) and no significant difference at 24 months (*p* = 0.446). Conversely, proteinuria was not significantly different at 6 and 12 months based on ABMR status (*p* = 0.585 and 0.414, respectively) but became significantly higher in ABMR patients at 24 months (median 305 vs. 147 mg/m^2^; *p* = 0.017) (Figure 2).

### 2.3. Incidence and Characterization of HLA Antibodies

Among the 92 patients with available antibody testing data at each timepoint, 26.1% developed nDSAs by 6 months, rising to 41.3% at 24 months. DSAs were detected in 18.5% of patients at 6 months and in 21.7% at 24 months. Children who were nDSA-positive at baseline had significantly lower body weight (Pearson’s r = −0.313, *p* < 0.001), and 34% of those who were nDSA-negative at baseline became positive in the first two years after transplantation, and 55% of those positive at baseline remained so over that period (McNemar’s χ^2^ = 33.8, *p* < 0.001). Combined positivity (DSA + nDSA) was observed at 11.9% at 6 months and reached 16.3% at 24 months (Figure 3). Multinomial logistic regression (McFadden’s R^2^ = 0.378, *p* < 0.001) identified ATG induction (odds ratio [OR] = 22.66, 95% confidence interval [CI]: 5.50–93.30, *p* < 0.001) and PVB19 plasma positivity (OR = 7.79, 95% CI: 1.25–48.62, *p* = 0.028) as predictors of nDSA formation, whereas BK polyomavirus (BKV) viremia was inversely associated (OR = 0.0328, 95% CI: 0.0017–0.620, *p* = 0.023). No other significant predictors of anti-HLA antibodies were identified. In particular, in a subgroup analysis, patients with immune-mediated CKD (e.g., glomerulopathies, HIV-associated nephropathy, lupus nephritis, IgA nephropathy; N = 30) had a nDSA prevalence of 21.6%, compared with 14.7% in those with non-immune-mediated CKD (CAKUT, ciliopathies, metabolic disease, or asphyxia; N = 62; χ^2^ = 1.34, *p* = 0.247).

No significant associations were found between the presence of nDSAs, DSAs, or their interaction and eGFR at 6, 12, or 24 months in the ANCOVA and ANOVA models (*p* > 0.3 for all comparisons). However, in non-parametric analyses stratified by antibody-mediated rejection (ABMR), eGFR was significantly lower at 6 months in patients who developed ABMR (*p* = 0.042), while proteinuria was significantly higher at 24 months (*p* = 0.017). A clear correlation was observed between the antibody titer and ABMR. Both nDSAs and DSAs showed significantly higher MFI values in ABMR-positive patients compared with ABMR-negative patients (5.203 vs. 1.428 for nDSAs; 4.931 vs. 1.260 for DSAs). This association was illustrated through separated boxplots, which demonstrate the distinct antibody intensity distributions according to ABMR status (Figure 4).

### 2.4. Histological Findings

A total of 272 biopsies were analyzed. The most frequent finding throughout the study period was Banff 1 (normal biopsy), detected in 55/92 (59.8%) at 6 months, 47/91 (51.6%) at 12 months, and 42/89 (47.2%) at 24 months, for a cumulative count of 144/272 (52.9%). Other categories, such as Banff 2 (antibody-mediated changes) and Banff 3 (borderline changes), remained relatively stable in frequency (each representing approximately 9–10% of total biopsies), while the proportion of Banff 5 (IFTA—interstitial fibrosis and tubular atrophy) diagnoses increased with time, from 0% at 6 months to 5.9% at 24 months. Similarly, Banff 6 (other changes) and “not adequate” samples were occasionally reported at all timepoints but without a clear temporal pattern. The distribution of histological findings is presented in Figure 5. Statistical analysis by the chi-squared test showed a significant association between Banff classification and biopsy timing (χ^2^ = 29.4, *p* = 0.003), suggesting that the histopathological profile of graft biopsies evolves during follow-up, with a notable decline in purely normal histology and a gradual emergence of chronic or borderline lesions.

### 2.5. Predictors of Antibody-Mediated Rejection

A multivariable Cox proportional hazards model was used to identify independent predictors of ABMR in the study cohort. The forest plot illustrates the hazard ratios (HRs) with 95% confidence intervals (CIs) for each variable on a logarithmic scale. Across the cohort, 9.8% of patients experienced ABMR at each of the three timepoints considered (6, 12, and 24 months). A binomial logistic regression model (McFadden’s R^2^ = 0.583, *p* < 0.001) revealed that PVB19 tissue positivity was an independent predictor of ABMR (OR = 9.53, 95% CI: 1.46–62.28, *p* = 0.019), as was EBV plasma positivity (OR = 20.53, 95% CI: 1.30–323.27, *p* = 0.032). The presence of both DSAs and nDSAs conferred a markedly higher risk of ABMR (OR = 78.39, 95% CI: 3.80–1616.95, *p* = 0.005), although isolated DSAs or nDSAs alone did not reach statistical significance in the same model. Furthermore, the presence of baseline nDSAs did not emerge as an independent predictor (OR 1.37, 95% CI 0.20–9.30, *p* = 0.747). Antibody titer, measured as MFI over 24 months, was also higher in ABMR patients for both nDSAs and DSAs. For independent sample Mann–Whitney tests, the median MFI of nDSAs was 5203 MFI in ABMR compared with 1428 MFI in non-ABMR patients (*p* = 0.002) and 4931 vs. 1260 MFI for DSAs (*p* = 0.004), although the distribution ranges were broad. The prognostic impact of anti-HLA antibodies and viral factors on ABMR over time was further analyzed by a Cox proportional hazards model. The presence of both DSAs and nDSAs yielded an HR of 45.10 (95% CI: 9.97–203.96, *p* < 0.001), while isolated nDSAs and DSAs conveyed HRs of 6.43 (95% CI: 1.28–32.39, *p* = 0.024) and 12.10 (95% CI: 1.32–110.84, *p* = 0.027), respectively, versus antibody-negative patients. Parvovirus B19 tissue positivity showed a significant association with ABMR (HR = 7.70, 95% CI: 2.72–21.80, *p* < 0.001) in the same time-to-event analysis, as far as the administration of steroids any other day compared with daily administration (HR = 9.89, 95% CI: 1.96–49.80, *p* = 0.005). Forest plots with a logarithmic scale for ABMR predictors are reported in Figure 6.

## 3. Discussion

The development of DSAs following kidney transplantation has been associated with poorer long-term graft survival and an increased risk of late antibody-mediated rejection (AMR) [9,11]. In addition to nDSAs, de novo non-donor-specific antibodies (dnnDSAs) may also emerge post-transplant; however, their clinical significance remains a matter of ongoing debate [12,13]. To better elucidate the impact of dnnDSAs on pediatric kidney transplant recipients, we carried out a post-transplant analysis in a pediatric cohort of 92 children.

In brief, our results confirm a progressive increase in antibody prevalence over the first two years post-transplantation, with nDSAs being more frequent than DSAs at all timepoints, providing new insights into the prevalence, dynamics, and clinical impact of non-donor-specific (nDSAs) anti-HLA antibodies in pediatric kidney transplantation.

In our cohort, the incidences of anti-HLA antibodies, DSAs, and nDSAs (respectively 21.7% and 41.3% at 24 months) are consistent with previous studies, which reported similar findings in pediatric populations. In fact, in a recent study on unsensitized pediatric kidney recipients, nDSAs were detected in 30% of patients within two years, while DSA incidence was around 24% [10]. Similarly, another study using data from the European CERTAIN registry reported DSA development in 20–25% of children during the first post-transplant year but did not focus on nDSAs [4]. Differently, Hourmant et al. analyzed 1229 adult kidney transplant recipients transplanted between 1997 and 2002 and reported post-transplant anti-HLA antibodies in 17% of patients (nDSAs in 11.2% and DSAs in 9.5% of patients), while the vast majority (83%) remained antibody-negative [9]. These discrepancies may reflect age-related differences in immune system maturation, HLA exposure, and immunosuppressive management, highlighting the distinct immunologic context of pediatric transplantation. Conversely, in our cohort, 38% of patients had detectable baseline nDSAs. This finding was associated with lower body weight (likely reflecting a higher burden of CKD-related complications in smaller infants) and with more severe clinical features, including higher proteinuria and reduced renal function [14]. However, baseline nDSA positivity alone does not independently predict ABMR in multivariable analyses, indicating that its presence does not fully capture the complexity of anti–HLA antibody evolution or rejection risk.

No patient died or experienced graft failure during follow-up in our cohort. Kidney function, as assessed by eGFR, remained stable over time, with median values of 75.0, 76.0, and 72.0 mL/min/1.73 m^2^ at 6, 12, and 24 months, respectively (*p* = 0.963). Previously published pediatric cohorts also reported stable short- to mid-term eGFR in low-immunological-risk recipients. In contrast to stable eGFR, proteinuria increased significantly over time (*p* = 0.013), reaching a median of 201 mg/m^2^ at 24 months. This progressive rise, especially in patients who developed ABMR (305 vs. 147 mg/m^2^; *p* = 0.017), supports the hypothesis that proteinuria may act as an early surrogate marker of graft inflammation. In fact, in a single-center study by Yilmaz et al., proteinuria was reported in 65% of pediatric transplant recipients with stable renal function, indicating that proteinuria may precede detectable eGFR decline [15]. Proteinuria has similarly been identified as an early sign of graft pathology in longitudinal studies, including the work by Naesens et al., which established proteinuria as a strong non-invasive marker of histological graft damage and predictor of long-term graft loss in adult recipients [16].

Importantly, the combined presence of DSAs and nDSAs in our cohort conferred a dramatically increased risk of ABMR (HR = 45.10; *p* < 0.001), with isolated nDSAs and DSAs also associated with increased risk (HR = 6.43 and 12.10, respectively). These findings are consistent with previous studies in adult recipients, which demonstrated a pathogenic role of nDSAs through cross-reactivity with donor HLA antigens or complement activation [9,17]. In our cohort, patients with ABMR had significantly higher median MFI values for both DSAs and nDSAs (5203 vs. 1428 for nDSAs; 4931 vs. 1260 for DSAs), supporting previous observations by Claisse et al. [18], who demonstrated that higher MFI values correlate with increased functional pathogenicity and complement-binding capacity, ultimately contributing to microvascular injury.

On the contrary, Cioni et al. reported no adverse impact of nDSAs on graft outcomes in unsensitized pediatric recipients [10]. However, the lower antibody titer and limited complement-binding capacity of nDSAs in that cohort compared with ours may explain the discrepancy. However, even in that study, histological AMR and microvascular inflammation were frequently observed in patients who developed both DSAs and nDSAs, supporting the idea of synergistic alloimmune injury when both antibody types are present and functionally active.

Viral infections also emerged as relevant cofactors in humoral alloimmunity in our study. EBV viremia and intrarenal Parvovirus B19 (PVB19) infection were both significantly associated with ABMR, and PVB19 also correlated with nDSA formation. These findings are in line with previous evidence implicating viral-mediated endothelial injury in antibody formation and rejection [19,20]. Furthermore, the association between EBV infection and humoral rejection is supported by increased plasma cell infiltrates and C4d deposition described by Aiello et al. [21]. A limitation of our study is the lack of baseline graft histology and virological screening (e.g., Parvovirus B19 PCR) at transplantation, which precludes one from assessing whether donor-derived viruses contributed to glomerular injury or nDSA formation, as previously reported [19].

## 4. Materials and Methods

### 4.1. Patients and Methods

This retrospective observational study included 92 consecutive pediatric kidney transplant recipients (64.1% male) who underwent transplantation between January 2015 and December 2022 at our Pediatric Kidney Transplant Center.

According to our protocol, immunosuppression is induced with basiliximab (10 mg for patients <30 kg; 20 mg for those >30 kg on post-operative days 0 and 4) for first transplants or anti-thymocyte globulin (ATG) for second or third transplants and maintained with a triple-drug immunosuppressive regimen including a calcineurin inhibitor (CNI, tacrolimus or cyclosporine), plus mycophenolate mofetil (600 mg/m^2^/day in two divided doses) or everolimus, and prednisone. Progressive reductions in CNI and steroids are applied from month 2+ to reach maintenance target levels or dosage, respectively.

Graft function was assessed at 6, 12, and 24 months after transplantation using serum creatinine, estimated glomerular filtration rate (eGFR) calculated with the revised Schwartz formula, and 24 h urinary protein excretion. Chronic kidney disease (CKD) staging was defined according to KDIGO guidelines [22]. Viral surveillance included routine monitoring for cytomegalovirus (CMV), Epstein–Barr virus (EBV), BK polyomavirus (BKV), Parvovirus B19, and adenovirus. Quantitative real-time PCR (qRT-PCR) assays were used to detect viral DNA in blood and in biopsy specimens using RT-PCR and considered in the outcome analysis when viral DNAemia was ≥1000 copies/mL.

Protocol kidney biopsies were performed at the same timepoints, regardless of graft function. All biopsies were reviewed by experienced renal pathologists and graded according to the 2017 Banff criteria [23]. Biopsy-proven acute cellular rejection episodes were treated with pulse intravenous methylprednisolone. Patients developing late AMR were treated with a protocol, including a combination of plasmapheresis, i.v. human Ig and anti-CD20 monoclonal antibody.

De novo anti-HLA antibodies were screened at 6, 12, and 24 months post-transplant using a Luminex-based single antigen assay (LABScreen^®^ Single Antigen, One Lambda, West Hills, CA, USA), with results expressed as mean fluorescence intensity (MFI). In a small subset of early transplants, anti-HLA screening was initially performed using ELISA-based assays prior to Luminex implementation. Given the lower sensitivity and specificity of ELISA [24], especially for class II antibodies, these results were excluded. All analyses in this study are based exclusively on Luminex data. To ensure accurate classification, all recipients and donors underwent high-resolution HLA typing at the HLA-A, -B, -C, -DRB1, and -DQB1 loci. Antibodies were classified as donor-specific (DSAs) when their specificity matched any mismatched HLA antigen present in the donor. Antibodies with no reactivity against donor antigens were defined as non-donor-specific (nDSAs), following current international consensus and expert recommendations [9]. A positivity threshold of mean fluorescence intensity (MFI) ≥ 2000 was applied, consistent with clinical practice and the prior literature. All results were interpreted by two experienced immunologists, and DSA assignment was confirmed through cross-referencing with donor HLA profiles.

Clinical, immunological, virological, and histopathological data were collected at each follow-up. Patients were stratified into four groups based on their post-transplant antibody profile: DSA only, nDSA only, both DSAs and nDSAs, or no anti-HLA antibodies. Comparative analyses were performed to evaluate differences in renal function, incidence and severity of rejection episodes, viral infections, and protocol biopsy findings. Adjustments to immunosuppressive regimens and other clinical management decisions were also recorded.

### 4.2. Statistical Analysis

Descriptive statistics were used to summarize clinical, biochemical, immunological, and virological data at baseline and during follow-up (6, 12, and 24 months post-transplantation). Continuous variables were expressed as mean ± standard deviation or median and interquartile range, as appropriate. Normality was assessed using the Shapiro–Wilk test. Group comparisons for continuous variables were conducted using either the independent samples t-test or the Mann–Whitney U test for non-normally distributed data. For comparisons among more than two timepoints, the Friedman test for repeated measures was employed, followed by Durbin–Conover pairwise comparisons. Differences in renal function (eGFR) and proteinuria across patient subgroups defined by anti-HLA antibody profiles (DSA only, nDSA only, both DSAs and nDSAs, or antibody-negative) were evaluated through ANCOVA models and one-way ANOVA or Kruskal–Wallis tests with post hoc Dwass–Steel–Critchlow–Fligner procedures, depending on data distribution. Correlation analyses between immunological, virological, and clinical variables were performed using Pearson’s correlation coefficients. Logistic regression models (binomial and multinomial) were used to identify factors independently associated with the presence of anti-HLA antibodies and antibody-mediated rejection (ABMR). Multivariable Cox proportional hazards models were employed to assess the impact of antibody development on graft survival. All analyses were conducted using Jamovi (version 2.6) [25] and R (version 4.4) [26], with statistical significance set at *p* < 0.05.

## 5. Conclusions

This study highlights the critical role of comprehensive immunological surveillance in pediatric kidney transplantation, emphasizing the need for monitoring both DSAs and nDSAs. While isolated nDSAs do not seem to correlate with immediate graft dysfunction, their coexistence with DSAs—particularly at high MFI—is strongly associated with ABMR, suggesting a synergistic alloimmune effect. Given the observed correlation between high MFI values and microvascular injury, MFI quantification and complement-binding assays should be incorporated into routine post-transplant evaluations to better stratify risk. Furthermore, the observed association between viral infections (especially EBV and Parvovirus B19) and both the development of nDSAs and ABMR underscores the importance of integrated virological and immunological monitoring, especially in children, who are more vulnerable to viral infections as they are often seronegative at the time of transplant and may receive a seropositive donor organ [21,22]. These findings support a risk-adapted approach to pediatric recipients and call for prospective multicenter studies to clarify the long-term impact of nDSAs, define clinically relevant MFI thresholds, and guide personalized therapeutic strategies to improve long-term outcomes.

## Figures and Tables

**Figure 1 ijms-26-05870-f001:**
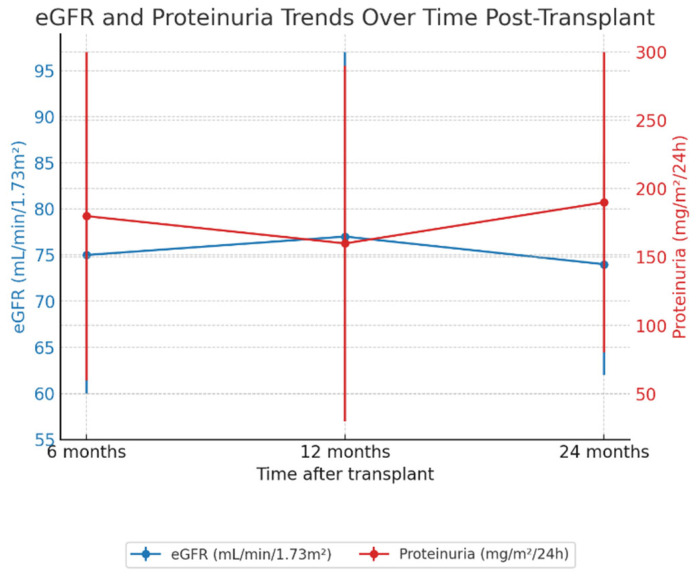
Trends in kidney function and proteinuria after pediatric kidney transplantation. Median estimated glomerular filtration rate (eGFR, blue line) and proteinuria (red line) at 6, 12, and 24 months post-transplant. Error bars represent interquartile ranges. While eGFR remained stable over time (*p* = 0.963), proteinuria significantly increased (*p* = 0.013).

**Figure 2 ijms-26-05870-f002:**
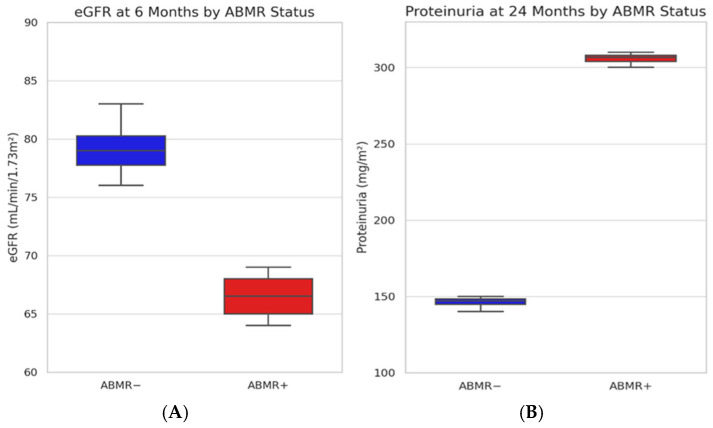
(**A**) Distribution of estimated glomerular filtration rate (eGFR) at 6 months and (**B**) proteinuria at 24 months post-transplantation, stratified by ABMR status. Boxplots show median, interquartile range, and individual values. Blue = ABMR negative; red = ABMR positive.

**Figure 3 ijms-26-05870-f003:**
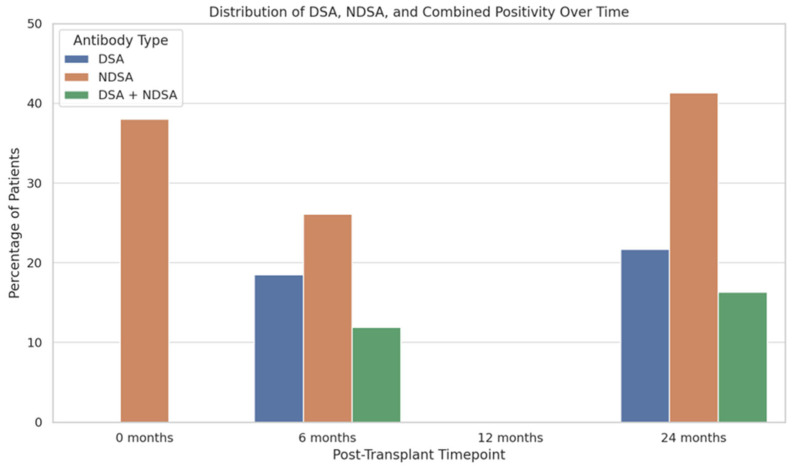
Bar chart showing the prevalence of non-donor specific antibodies (nDSAs), donor-specific antibodies (DSAs), and combined positivity at baseline, 6, and 24 months post-transplant among 92 patients with available antibody testing. Values are reported as percentages.

**Figure 4 ijms-26-05870-f004:**
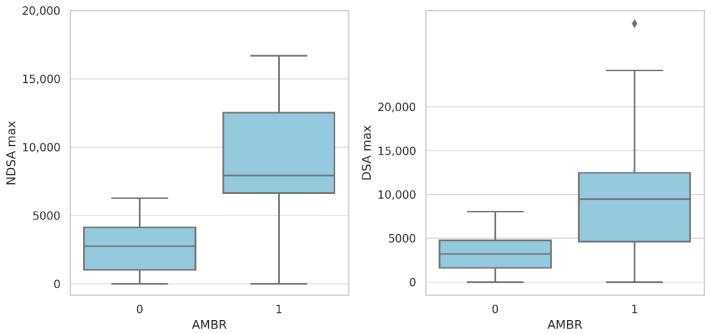
Boxplots showing the maximum mean fluorescence intensity (MFI) values for non-donor specific anti-HLA antibodies (nDSAs, **left**) and donor-specific antibodies (DSAs, **right**), stratified by the presence of antibody-mediated rejection (ABMR). Higher MFI levels were observed in ABMR-positive patients, suggesting a potential association between antibody strength and the risk of humoral rejection.

**Figure 5 ijms-26-05870-f005:**
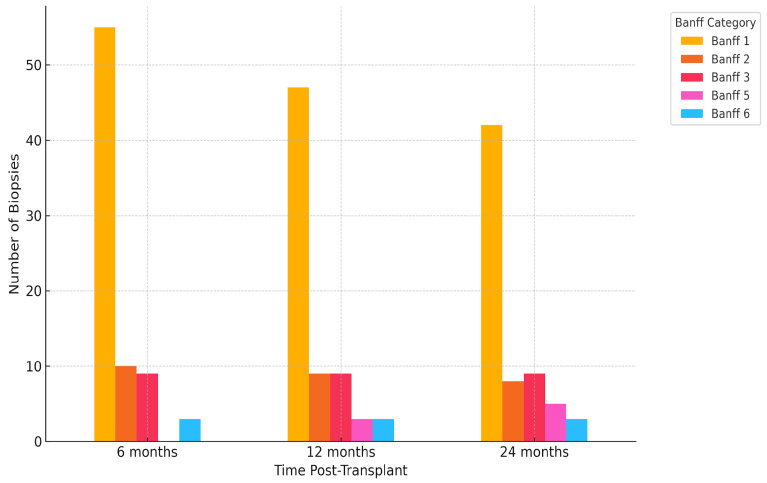
Distribution of Banff categories at 6, 12, and 24 months post-transplantation. The figure illustrates the temporal evolution of biopsy findings according to the Banff classification system.

**Figure 6 ijms-26-05870-f006:**
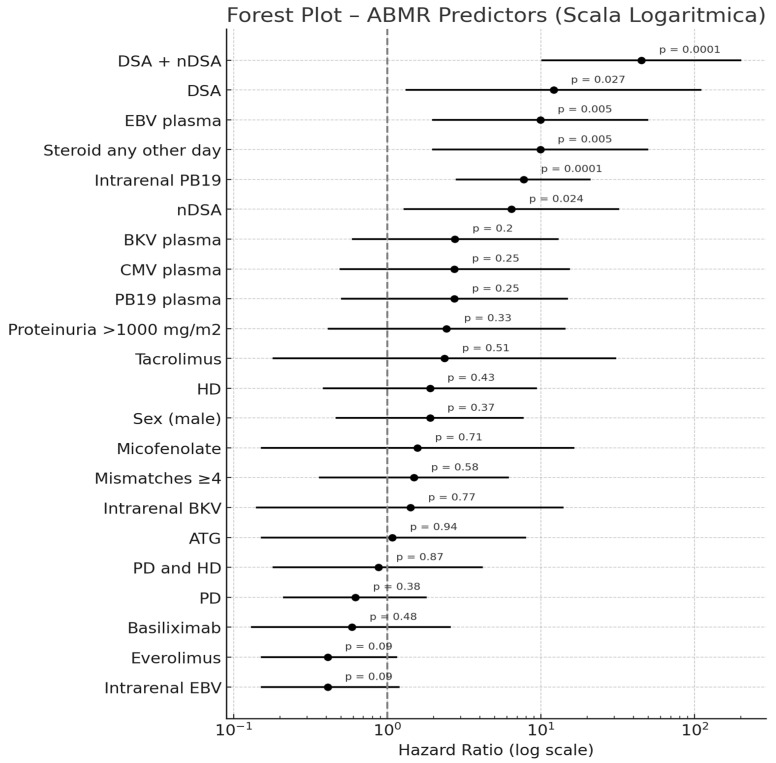
Forest plot in logarithmic scale for ABMR predictors.

**Table 1 ijms-26-05870-t001:** Demographic and clinical characteristics of the study population.

CHARACTERISTICS		
Male sex (%)	59	64
Age at transplant, mean (IQR)	11.5	2–21
Weight at transplant kg, mean (IQR)	26.5	7–83
Mismatch, mean (IQR)	4	1–6
First transplant, %	82	89
PD, %	38	41
HD, %	21	22
PD + HD, %	15	16
No dialysis, %	18	19
1st transplant, %	82	89
2nd transplant, %	9	9
3rd transplant, %	1	1
Deceased donor, %	68	73
Baseline nDSAs, %	38	
Cause of ESKD, %		
CAKUT	44	47
glomerulopathy	19	20
ciliopathy	15	16
unknown	4	4
inborn error of metabolism	3	3
chronic TIN	3	3
perinatal asphyxia	3	3
HIV related	1	1
Immunosuppressive induction, %		
basiliximab	84	91
ATG	13	14
tacrolimus	49	53
mycophenolate mofetil	80	86
steroid	92	100
ciclosporin	38	41

Demographic and clinical characteristics of pediatric kidney transplant recipients were included in this study. Values are expressed as percentages or as mean (interquartile range, IQR) where indicated. Abbreviations: CAKUT, congenital anomalies of the kidney and urinary tract; PD, peritoneal dialysis; HD, hemodialysis; nDSAs, non-donor-specific antibodies.

## Data Availability

Data are contained within the article; further inquiries can be directed to the corresponding author.

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
