# Peer review of "The Impact of Non-Donor-Specific HLA Antibodies on Antibody-Mediated Rejection in Pediatric Kidney Transplant Recipients"

_ijms, 2025, doi:10.3390/ijms26125870_

Round 1

Reviewer 1 Report

Comments and Suggestions for Authors

Dear Authors,

This is an interesting study focusing on non-donor-specific anti-HLA antbodies (nDSA) in pediatric KTx recipients, however some major and minor conerns can be raised, which needs major revision.

Major concerns:

1.) The authors do not present data on pretransplant (preTx) nDSA. We do not know, whether any nDSA existed before / at the time of transplantation, thus it is not clear, whether nDSA presented in the paper are preTx or de novo! This is a major point, and should be cleared, by each patient individually, and all result should be reevaluated based on this.

2.) The native kidney disease of the patients was glomerulopathy in 20,6%. These diseases e.g. (or others like HIV related, etc.) may have (auto)immune background which might provoke nDSA formation. I suggest separate evaluation of the result:

  • diseases, where nDSA formation is triggered by the original disease
  • CAKUT group (large enough) AND/OR ciliopathies

3.) We need data on the immunosuppression of the ABMR pos. and neg. within nDSA groups prior the diagnosis of ABMR. Was there any difference? I.e. weaker immunosuppression leads to more ABMR positivity caused by nDSA?

4.) Viruses - e.g. parvovirus B19 - may per se be the cause of glomerulonephritis and induce antibody production. The native kidney disease and virus status should be checked for simultaneous presence to clear the possible causative relationship and for presence of nDSA.

Minor concerns:

1.) Presentation of the data should be improved:

  • in Fig. 2. use the same colour for ABMR pos. and neg. boxplots
  • in all Figs show all time points' data (0-6-12-24 months) to have a coherent view on all parameters with the time flow - even if there are no significant changes - this is especially important in Fig. 3 for DSA data!

2.) Use abbreviations consequently: 

  • PB19 vs. PVB19
  • nDSA vs. NDSA

3.) The abstract says: 92 first-time Tx pts vs. results says: 89,1% 1st, 9,8% 2nd, 1,1% 3rd Tx

Reviewer 2 Report

Comments and Suggestions for Authors

Sangermano et al., performed a retrospective study to gain a better understanding of the emergence and clinical impact of anti-HLA antibodies in pediatric kidney recipients.  They assessed the frequency of DSA and non-donor-DSA antibody formation after transplantation and investigated the association between antibody development and clinical outcomes such as graft function, rejection episodes, and viral infections.  Their conclusion is that there is potential prognostic value of anti-HLA-non-donor-specific antibodies, which currently have unclear significance.

Critics

Gaining a better understanding of which DSA and nDSA antibodies are present would be of value.  Basically, were there class I and/or II antibodies detected and did they account for cross-reactivity etc.

More clarity on how they are "calling" DSA vs non-DSA is critical.  Providing a supplementary data with DSA vs non-DSA is an important factor in assessing the validity of their study.

Are clinical teams over-immunosuppressing patients based on their sensitization status even in the absence of donor specific HLA antibodies- leaving the patient's more susceptible to infections?

Including the etiology of disease (e.g., immune-mediated vs genetic) is a variable that should be accounted for especially when discussing how a pediatric population differs to adult population.

More information on the ELISA based method is necessary- not clear on what was done using this approach.  Currently, routine HLA antibody detection is done using Luminex approaches.

Round 2

Reviewer 1 Report

Comments and Suggestions for Authors

Thank you, I accept.

Reviewer 2 Report

Comments and Suggestions for Authors

Overall, the authors responded to the critics.